# Presence of Induced Weak Ferromagnetism in Fe-Substituted YFe_x_Cr_1−x_O_3_ Crystalline Compounds

**DOI:** 10.3390/nano12193516

**Published:** 2022-10-08

**Authors:** Roberto Salazar-Rodriguez, Domingo Aliaga Guerra, Jean-Marc Greneche, Keith M. Taddei, Noemi-Raquel Checca-Huaman, Edson C. Passamani, Juan A. Ramos-Guivar

**Affiliations:** 1Facultad de Ciencias, Universidad Nacional de Ingeniería (UNI), Av. Túpac Amaru 210, Rímac, Lima 15333, Peru; 2Institut des Molécules and Matériaux du Mans (IMMM UMR CNRS 6283), University Le Mans, Avenue Olivier Messiaen, Cedex 9, 72085 Le Mans, France; 3Oak Ridge National Laboratory (ORNL), Oak Ridge, TN 37831, USA; 4Centro Brasileiro de Pesquisas Físicas (CBPF), R. Xavier Sigaud, 150, Urca, Rio de Janeiro 22290-180, Brazil; 5Departamento de Física, Universidade Federal do Espírito Santo, Vitória 29075-910, Brazil; 6Grupo de Investigación de Nanotecnología Aplicada para Biorremediación Ambiental, Energía, Biomedicina y Agricultura (NANOTECH), Facultad de Ciencias Físicas, Universidad Nacional Mayor de San Marcos, Av. Venezuela Cdra 34 S/N, Ciudad Universitaria, Lima 15081, Peru

**Keywords:** DM interaction, crystalline YFeO_3_, magnetic properties, enhanced weak ferromagnetism, exchange interactions

## Abstract

Fe-substituted YFe_x_Cr_1−x_O_3_ crystalline compounds show promising magnetic and multiferroic properties. Here we report the synthesis and characterization of several compositions from this series. Using the autocombustion route, various compositions (x = 0.25, 0.50, 0.6, 0.75, 0.9, and 1) were synthesized as high-quality crystalline powders. In order to obtain microscopic and atomic information about their structure and magnetism, characterization was performed using room temperature X-ray diffraction and energy dispersion analysis as well as temperature-dependent neutron diffraction, magnetometry, and ^57^Fe Mössbauer spectrometry. Rietveld analysis of the diffraction data revealed a crystallite size of 84 (8) nm for YFeO_3_, while energy dispersion analysis indicated compositions close to the nominal compositions. The magnetic results suggested an enhancement of the weak ferromagnetism for the YFeO_3_ phase due to two contributions. First, a high magnetocrystalline anisotropy was associated with the crystalline character that favored a unique high canting angle of the antiferromagnetic phase (13°), as indicated by the neutron diffraction analysis. This was also evidenced by the high magnetic hysteresis curves up to 90 kOe by a remarkable high critical coercivity value of 46.7 kOe at room temperature. Second, the Dzyaloshinskii–Moriya interactions between homogenous and heterogeneous magnetic pairs resulted from the inhomogeneous distribution of Fe^3+^ and Cr^3+^ ions, as indicated by ^57^Fe Mössbauer studies. Together, these results point to new methods of controlling the magnetic properties of these materials.

## 1. Introduction

Bulk orthoferrites and orthochromites have been the subject of various studies since the 1950s, but the interest of the scientific community has recently been renewed due to their possible technological applications in sensors, switching devices, and spintronics [1,2]. New research is focused on improving novel synthesis methods, consequently producing materials with different and improved magnetic and electrical properties. It is important first to point out that the first publications were carried out almost exclusively with single crystal samples, a synthesis process that requires specialized equipment not often found in ordinary laboratories, and that inherently prevents the production of large quantities of materials [3,4]. Currently, new synthesis methods have been explored to produce polycrystalline samples that have the advantage of rapid preparation, low cost, and the ability to produce relatively large masses of material [4]. On the other hand, developments in nanotechnology have also made it possible to obtain nanocrystalline orthoferrites with highly controlled stoichiometry using wet methods, such as the sol–gel approach, allowing continuous doping between the orthoferrite and orthochromite endmembers [5]. Such synthesis developments may be instrumental in studying the Fe–Cr phase diagram, which has been historically challenging due to the thermodynamic considerations of the Y_2_O_3_–Fe_2_O_3_ binary phase diagram, in which magnetite or ternary garnets can easily be obtained as parasitic (or secondary) phases. Therefore, the use of these new techniques to scale up the synthesis and carefully control the composition allows unprecedented access to the Fe–Cr phase diagram, potentially enabling the optimization and understanding of the interesting magnetic properties of these materials.

The YFeO_3_ orthoferrite (YFO) and the YCrO_3_ orthochromite (YCO) compounds crystallize in the Pnma centrosymmetric space group symmetry and are biferroic with high Néel temperatures (*T_N_*). Specifically, these materials are antiferromagnetic with *T_N_* = 644 K and 140 K for YFO and YCO, respectively, with the a-direction of the unit cell as the easy magnetization axis. Moreover, these materials exhibit weak ferromagnetism, which has been attributed to the canting of magnetic moments along the c-direction, leading to the convincing observation of ferromagnetic behavior [5]. As early as the 1960s, Treves [6] proposed an antisymmetric interaction mechanism (Dzyaloshinskii–Moriya (DM) exchange) for this type of ferromagnetism in orthoferrites based on torque measurements in single crystals of rare earth orthoferrites (RFeO_3_, R = rare earth and Y). Considering these intriguing physical properties and the possibility to combine the two above materials (i.e., forming a Fe-substituted YCO phase), improved electrical and magnetic properties are expected. In particular, the case of the YFe_0.5_Cr_0.5_O_3_ compound is very interesting because it has the largest magnetoelectric effect in the series [7] and may present a spin reorientation phenomenon according to the literature [8]. However, most of these studies have been performed on bulk polycrystalline samples, leaving the effects of another tuning parameter, namely, the crystallite size, unstudied. To optimize and understand the magnetoelectric effect in these materials, one could use the crystalline size as well as the application of large magnetic fields to try to stabilize different magnetic configurations or tune the canting angle and thus potentially improve the magnetoelectric effect.

In this work, the structural and magnetic properties of the Fe-substituted YCO phase (i.e., theYFe_x_Cr_1−x_O_3_ compounds) in the crystalline regime are studied in detail. The structural and composition features were obtained from Rietveld refinements and scanning electron microscopy (SEM), which showed (i) a single phase in the studied compounds and (ii) their nanoscale characteristics. The magnetic properties of the Fe-substituted compounds were studied by performing direct current magnetization measurements at and between 300 K (room temperature (RT)) and 5 K using both zero-field-cooling (ZFC) and field-cooling (FC) protocols, while the local magnetic properties were investigated using ^57^Fe Mössbauer spectrometry performed at 300 K and 77 K and under an external magnetic field. The obtained results suggested an enhancement of WFM associated mainly with the sub-micrometric size character of the YFO phase (84 (8) nm), which has favored a relevant spin canting of 13°, i.e., a net spin contribution related to the sample finite-size effect. This property is not observed in the single and polycrystalline systems reported in the literature. In addition, the presence of the finite-size effect (spin-canting) in our sample was also reflected in the increase of the saturation magnetization that reached 0.79 emu/g at 3.5 kOe.

## 2. Materials and Methods

The whole series of YFe_x_Cr_1−x_O_3_ perovskites was synthesized by the combustion method by stoichiometrically mixing the following initial reactants: Y(NO_3_)·6H_2_O, Fe(NO_3_)_3_·9H_2_O, Cr (NO_3_)_3_·9H_2_O, urea, and glycine. To improve the crystallization process, all powder samples were heated up to 1200 °C and annealed under ambient conditions. The synthesized samples were labeled as the RSx series, where RS1, RS2, RS3, RS4, RS5, RS6, and RS7 correspond to x = 0, 0.25, 0.50, 0.60, 0.75, 0.90, and 1.0, respectively.

Structural characterization was carried out using Bruker Advance D8 X- ray diffraction equipment (Bruker Corporation, Billerica, MA, USA), operating with a Cu–Kα radiation source (1.5418 Å wavelength), and the X-ray diffraction (XRD) diffractograms were recorded at RT with a 2θ from 15° to 65° in a step of 0.02° and with an accumulating time of 10 s. Neutron diffraction (NPD) of the RS3 (YFe_0.5_Cr_0.5_O_3_) sample was performed on the HB-2A line of the High Flux Isotope Reactor (HFIR) at Oak Ridge National Laboratory (ORNL) [9]. X-ray and NPD data were analyzed using the FullProf suite (Gif sur Yvette Cedex, France, version January 2021). In all nuclear diffraction peak modeling, the previously reported orthorhombic crystal structure (space group Pnma) was found to account for all observed peak positions and intensities, identified using the software Match v3. As initial cell parameter values, we employed a = 5.59 Å, b = 7.59 Å, and c = 5.27 Å (Match entry 210–1387) and allowed the parameters to refine during the profile fitting for the different temperatures and compositions. The instrumental resolution function (IRF) of the X-ray diffractometer was obtained from the aluminum oxide (Al_2_O_3_) standard with Caglioti parameters: U = 0.0093, V = −0.0051, and W = 0.0013 [10]. The morphology, size, and composition of the powders were obtained using a TESCAN LYRA3 high-resolution scanning electron microscope (Tescan Brno s.r.o., Brno, Czech Republic) with an FEG type electron source coupled with an Oxford energy-dispersive X-ray spectroscopy (EDS) detector. Secondary electron imaging and atomic element mapping were acquired simultaneously using an accelerating voltage of 15 kV and a working distance of 9 mm.

Zero-field-cooling (ZFC) magnetic hysteresis loops (*M(H)* loops) were recorded at 300 K and 5 K using the vibrating sample magnetometer (VSM) option operating in a Dynacool (Quantum Design North America, San Diego, CA, USA) setup for a maximum applied field of 90 kOe. ZFC and warm-field-cooling (WFC) magnetization measurements, *M(T)*, were performed under two different probe fields: 50 Oe and 1000 Oe.

^57^Fe Mössbauer spectra were obtained with WissEl equipment (WissEl—Wissenschaftliche Elektronik GmbH, Starnberg, Germany) in a transmission geometry using a ^57^Co source diffused into an Rh matrix with an activity of about 1.5 GBq and mounted on a conventional constant acceleration vibrating electromagnetic transducer. The sample was in the form of a powder layer containing about 5 mg Fe/cm^2^. Spectra were obtained at 300 K and at 77 K in a bath cryostat. A thin foil of α-Fe was used at 300 K for calibration of the spectrometry (isomer shift values are given relative to Fe at 300 K). The modeling of the hyperfine structures was performed using a homemade Mosfit program based on the least squares method, and magnetic and quadrupolar components were composed of Lorentzian peaks.

## 3. Results and Discussion

### 3.1. XRD and Rietveld Analysis

From the Rietveld refinement of the XRD and neutron diffraction powder (NPD) patterns measured at RT and 2 K, respectively, the results suggested for all samples the presence of only an orthorhombic Pnma (No. 62) crystal structure [11], i.e., no secondary phase was observed. In addition, both experiments showed similar behavior for the lattice parameters a, b, and c within their uncertainties. Thus, the refined values of the cell parameters are plotted in Figure 1a,b, which suggested: (i) a linear evolution of the cell parameters (Figure 1a) at RT and 2 K as a function of iron concentration (x), (ii) the c parameter changed more rapidly than the a or b parameters, and (iii) a continuous increase in cell volume with Fe concentration (Figure 1b) at both temperatures. For the size estimation of YFeO_3_ crystallites (RS7 sample) (see refined diffractogram in Figure 1c), we used the modified Scherrer’s formula that expresses the anisotropic size broadening as a linear combination of spherical harmonics (SHP) if the anisotropic size contribution belongs only to the Lorentzian component of the total Voigt function [10]. Therefore, the explicit formula for the SPH approach of size broadening is given by Equation (1) [12]:(1)$$\beta_{h}=\frac{\lambda }{D_{h}cos\theta }=\frac{\lambda }{cos\theta }\sum_{lmp}a_{lmp}y_{lmp}(\Theta_{h},\;\Phi_{h})$$
where *h* is assigned to the (hkl) indices, $\beta_{h}$ is the size contribution to the integral width of reflection (hkl), *y_lmp_*(Θ_h_, Φ_h_) are the real components of spherical harmonics (arguments Θ_h_ and Φ_h_ are the polar and azimuthal angles of vector (hkl) with respect to a Cartesian crystallographic frame), and *a_lmp_* are the refined coefficients, related to the Laue class [13]. For the RS7 sample, a -1 Laue class was used.

The obtained profile refinement gave acceptable statistical parameters for the reliability factor, R_p_ (%); weighted profile residual, Rw_p_ (%); expected profile residual, R_exp_(%); and goodness of fit, χ^2^; having quantitative values as follows: R_p_ = 16.2%, R_wp_ = 10.4%, R_exp_ = 26.19%, and χ^2^ = 0.94, while the refined harmonic coefficients were found equal to Y_00_ = −0.00574, Y_20_ = 0.01395, Y_21_^+^ = −0.06448, Y_21_^−^ = 0.01518, Y_22_^+^ = 0.07557, and Y_22_^−^ = −0.03690, respectively. With these values, the anisotropic Lorentzian size broadening gave a mean crystallite value of 84 (8) nm. Hence, combining all above data, it can be inferred that the autocombustion method allowed nanocrystalline Fe-substituted YCrO_3_ powders with a single phase, orthorhombic-like structure to be obtained.

### 3.2. SEM Analysis

For all samples, the autocombustion method with final annealing up to 1200 °C yielded a distribution of agglomerates about (~200–500 nm) that form a series of interconnected chains, as typical found in autocombustion synthesis [4]. In Figure 2a–q, the RSx series shows similar morphologies as those obtained by Zhang et al. [4,14]. In all of them, the notorious polycrystalline nature can be observed. The STEM images and elemental analyses given by yellow, red, blue, and green colors (Figure 2d–m) show the evolution of the systems when the Fe concentration increases. The systematic formation of the Fe-substituted YCrO_3_ phase is noted in the EDS pattern given in Figure 2r, and the atomic percentage contribution is summarized in Table 1. We can roughly say that the particles produced by this combustion method have a similar morphology and similar dispersion in all tested concentrations. In addition, considering the uncertainties of the element contents in the samples, we can also affirm that Y and O are quite constant, while Fe increases and Cr decreases its contribution; this indicates that Fe enters the crystalline cell, due to the larger ionic radius of Fe^3+^ (0.645 Ӑ) (compared to Cr^3+^ = 0.615 Ӑ), which is consistent with the unit cell volume determined from X-ray and neutron diffractions, as previously discussed.

### 3.3. VSM Analysis

Figure 3 and Figure 4 illustrate the ZFC ± 90 kOe *M(H)* curves of the YFe_x_Cr_1−x_O_3_ series taken at 5 K (top) and 300 K (bottom), respectively. At RT, the samples with x = 0.25 and 0.50 behaved as ordinary paramagnets (see Figure 3d,f), while the samples with x = 0.60, 0.75, and 0.90 suggested an onset of a weak ferromagnetism (see Figure 4b,d,f). For the x = 1.0 sample (YFeO_3_ compound, RS7), shown in Figure 5a, *M(H)*, curves recorded at different temperatures show the magnetic features of a weak ferromagnet with high magnetic anisotropy, i.e., with characteristics similar to that found in the pure YFeO_3_ compound (set-like *M(H)* curve). Therefore, this sample (RS7) revealed an interesting and complex magnetic behavior that is mainly attributed in the literature to an exchange spring effect. In particular, the magnetic spring effect can be often observed by an exchange magnetic coupling between coexisting and interacting soft and hard magnets in a sample, as reported by Popkov et al. [5].

Thus, since the NDP, Rietveld, and SEM data of the SR7 sample suggested the presence of a single-phase structure, and the presence of two magnetic phases of two crystalline structures (as occurs in bilayer films) cannot be inferred as a reason for the observed phenomenon. However, the atomic disordering in the orthorhombic structure and the change in cell volume could lead to local different magnetic phases, which will be magnetically interacting and producing the observed *M(H)* behavior discussed above.

Looking at the 5 K *M(H)* loop for the x = 0 sample (see Figure 3a), we can observe the characteristic *M(H)* curve reported for the YCrO_3_ compound [15]. Below *T_N_,* the non-saturation regime of the *M(H)* curve occurred till values of +90 kOe, indicating a remarkable antiferromagnetic state, while above *T_N_,* a paramagnetic-like behavior was regarded; see Figure 3b. On the other hand, the loss of hysteresis in the RS2 and RS6 samples (Figure 3c,e and Figure 4a,c,e) indicated the substitution of Cr by Fe atoms in the orthorhombic crystal configuration, as confirmed by our XRD data. Table 2 contains the remanence (*M_r_*), coercivity (*H_C_*), and saturation magnetization (*σ_sat_*) values of the hysteresis ferromagnetic part. Using the slope of the *M(H)* curves, it was possible to subtract the antiferromagnetic contribution of the *M(H)* curves, thus leaving the purely ferromagnetic component, as shown in Figure 5b. Thus, these *M(H)* curves recorded at different temperatures really showed large *H_C_* fields, but their value decreased when the temperature decreased, concomitantly with the increase of the saturation magnetization (the area inside the *M(H)* loop remained nearly constant). In addition, all *M(H)* curves show more clearly the step-like behavior near the zero-applied field region of the *M(H)* curve, a feature discussed above and attributed to a magnetic spring-like effect.

The magnetic parameters in Table 2 were plotted as a function of Fe concentration (x), as seen in Figure 6a,b,d. At 300 K, the *H_C_* field dependence with x had two marked regions (I and II): (i) region-I can be interpreted as the magnetic domain reorientations (magnetization reversal) due to the increasing concentration of Fe atoms that are replacing Cr, forming the pure YFeO_3_ crystalline phase; (ii) region-II has relatively high values of the *H_C_* fields and that occur above x = 0.75, reaching a maximum value of 46.7 kOe for x = 1 (RS7 sample).

These values were larger than those reported by Popkov et al. [5] for four YFeO_3_ crystalline samples synthesized by different routes. On the other hand, the *M_r_* and *σ_sat_* values had a similar dependence with Fe concentration at 300 K and 5 K, as can be seen in Figure 6b,d. The behavior of *M_r_* vs. *σ_sat_* is shown in Figure 6c. The *M_r_* and *σ_sat_* quantities could reach maximum values of 0.75 and 0.79 emu/g, respectively. This *σ_sat_* value of 0.79 emu/g was consistent with others found in the literature for either powder or single crystals [3,4,16,17,18,19], as summarized in Figure 7. In particular, the value of 0.79 emu/g, obtained for a field of 3.5 kOe, was almost two times higher than the values reported by Zhang et al. [4] and four times higher than that obtained by Shen et al. [20] for a similar system. Therefore, the RS7 sample behaved as an ordinary single crystal of the YFO phase with a multidomain magnetic structure [4,20]. In addition, it is worth mentioning that the RS7 sample exhibited weak ferromagnetism enhanced at 90 kOe.

The WFC and ZFC *M(T)* measurements for all RSx samples were collected under two probe fields, namely, 50 Oe and 1000 Oe, and the results are shown in Figure 8, Figure 9 and Figure 10. For the lowest applied field, the ZFC and WFC *M(T)* curves, displayed in Figure 8a, clearly show the magnetization transition from the AFM to PM state of the YCrO_3_ compound at 159 K, assigned to *T_N_*. No other magnetic transition was observed in *M(T)* curves, indicating that no secondary phase was formed during the auto combustion synthesis, in agreement with the XRD data. For the YFe_0.25_Cr_0.75_O_3_ compound, the *T_N_* value increased to 174 K (see Figure 8b), but a further increase of Fe content, for example, x = 0.50, led to a cancelation of total magnetization and a compensation temperature between the antiferromagnetic sub-lattices of 245 K. The zero-net magnetization was observed as an enhancement of the diamagnetism contribution, as shown in Figure 8c. At x = 0.60 and 0.75, see Figure 8d,e, a slight increase in the magnetization was observed, in agreement with the onset of WFM, as also seen in the *M(H)* curves. At x = 0.90 and 1.0 (Figure 8f and Figure 9), a significant increase in the magnetization was observed with significant overlap between ZFC and WFC *M(T)* curves above 250 K.

The determination of the *T_N_* of the Fe-substituted YCO compounds was done recording the ZFC and WFC *M(T)* curves at a higher field (1000 Oe). At this probe field, the magnetization of the samples with x = 0.75 and x = 0.90 showed a strong interaction with the external field, confirming the enhancement of the WFM. From ca. 5 K to higher temperatures, both ZFC and WFC *M(T)* curves coincided for the sample with x = 0.9.

Based on the above experimental results, it can be inferred that the anisotropic exchange-spring in crystalline compounds cause a significant increase in the coercive field of 46.7 kOe at 300 K. This interesting magnetic response has also been observed by Popkov et al. [5]. In our case, the hard and soft magnetic phases are intrinsically correlated to the same structure, but they are due to chemical disorders in the sites of the orthorhombic crystal nanostructure. Moreover, the hysteresis loop shape depends on the finite-size effects under an applied DC magnetic field (in our case, we use the highest value reported in the literature of 90 kOe). Hence, the observed ascending/descending hysteresis loops at several temperatures is explained due to spin reorientation of the antiferromagnetic vector in the *x–z* plane, reaching the *z*-axis at a critical magnetic field, as reported by Jacobs et al. [21], where a value of 74 kOe at 4.2 K was obtained for the YFeO_3_ single crystal. According to Popkov et al. [5], in nanocrystalline materials, the typical WFM hysteresis cycle is observed only for the YFO phase when their grain sizes are equal and larger than 41 nm, i.e., the YFO material may exhibit WFM, and the exchange spring-like effect may occur due to its high magnetocrystalline anisotropy energy. Consequently, considering our experimental results that showed a grain size of 84 (8) nm, we can also expect the observed ascending/descending branch behaviors of the *M(H)* loops of the YFO sample. More precisely, the combined magnetic effects of the enhanced WFM and the presence of AFM interactions among the Fe ions of the different sites of the orthorhombic crystal structure gave rise to different local anisotropy contributions, producing high magnetocrystalline anisotropy due to the size effect and the enhancement of DM (Dzyaloshinskii–Moriya) interactions in the samples.

The two effects cannot be separated, and the improvement of WFM features can be explained assuming a canting angle of 13°, as demonstrated by previous neutron diffraction analysis [11]. The presence of AFM interactions in the Fe-substituted YCO compounds is also confirmed by the changes of *T_N_* values as a function of Fe content, as displayed in Figure 11. Indeed, the *T_N_* values increase nonlinearly with increasing Fe content, reaching the reference value for YFO [8]. Of course, the Fe substitution phenomenon is randomly changing locally the anisotropy by changing the lattice parameters, as shown by the XRD results. These modifications favor the spin reorientation and magnetization reversal phenomena.

### 3.4. Mössbauer Analysis

#### 3.4.1. Measurements at 300 K

In agreement with the magnetization data, the 300 K ^57^Fe Mössbauer spectra recorded for samples with different Fe concentrations show, on the one hand, partial (x = 0.50) or total (x = 0.25) paramagnetic behavior (see Figure 12a). On the other hand, the 300 K ^57^Fe Mössbauer spectra of the samples with x = 0.75 and 1.0 show six absorption lines due to the nuclear Zeeman interaction with a local magnetic hyperfine field (B_hf_). The refined values of the corresponding hyperfine parameters are given in Table 3. The values of isomer shift are typical of the presence of Fe^3+^ ions. Another important feature that should be highlighted is that the line widths of the Mössbauer spectra are generally broader for samples with x = 0.25, 0.50, and 0.75 compared with those of the RS7 sample (x = 1.0), the latter being expected to show less atomic disorder. Therefore, the broadening effect of magnetic lines is probably caused by different iron environments, since in the orthorhombic crystal structure of these perovskites, a 3d^5^ Fe^3+^ ion is usually surrounded by 2, 3, 4, 5, or 6 Cr^3+^ ions in octahedral sites. The result of the chemical disorder is a hyperfine magnetic field distribution, i.e., a distribution of static sextets. In particular, the fit of the ^57^Fe Mössbauer spectrum of the sample with x = 0.50 was done with two magnetic sextets and one quadrupolar doublet. The two sextets will represent the different local Fe environments of the orthorhombic crystal structure, while the doublet, best seen in the inset spectrum recorded in a low-velocity range, must be associated with Fe^3+^ ions in the paramagnetic state resulting from Cr^3+^-rich environments (*T_N_* < 300 K, e.g., for the x = 0.25, *T_N_* = 153 K). The features discussed above tell us that the Fe substitution is not homogeneous, leading to an assembly of clusters with different compositions, i.e., a chemical disorder in the octahedral sites (B-sites) of the orthorhombic crystal structure. Thus, the largest magnetic component can be attributed to Fe^3+^ ions preferentially surrounded by Fe^3+^ ions (*T_N_* > 300 K), while the quadrupolar doublet is associated with a neighborhood rich in Cr^3+^ ions, of course, with *T_N_* values lower than 300 K, as shown by our magnetization data.

#### 3.4.2. Measurements at 77 K

To better understand the local environment of the ^57^Fe ions in the orthorhombic crystal structure, additional measurements were made at 77 K for all Fe-substituted series, and the corresponding Mössbauer spectra are shown in Figure 12b. At 77 K, below the *T_N_* values of the Fe-substituted YCO compounds (see Figure 11), one would therefore expect a pure Zeeman nuclear interaction in all ^57^Fe spectra. The spectra, in general, show all six expected absorption lines, but with different broadening and asymmetries depending on the Fe content. While the 77 K Mössbauer spectrum of the YFO (x = 1) compound can be perfectly described by a single magnetic component, those of RS5, RS3, and RS2 require at least two magnetic components. The refined values of the hyperfine parameters are given in Table 3. Thus, to fit these 77 K spectra, we have two magnetic sextets to account for, at least, two octahedral configurations of Fe^3+^ ions for samples with non-zero x. The results clearly show that B_hf_ values decrease with increasing content. Even at 77 K, the spectrum of the YFe_.25_Cr_.75_O_3_ sample required an additional quadrupolar doublet, with a fraction of 5% of total spectra. Thus, considering that the magnetization data show a *T_N_* value for this sample equal to 153 K, the quadrupolar doublet must be associated with Fe^3+^ ions with a Cr^3+^ ion-rich neighborhood.

According to the Néel temperature of the series, it is understandable why the samples with x = 1.0 and 0.75 show hysteresis cycles, although the YFeO_3_ sample has a lower magnetic energy than the 50% sample. Similarly, for the other two samples with x = 0.25 and zero, the magnetic susceptibility is consistent with paramagnetic behavior, which is understandable due to their lower *T_N_* values than 300 K. The series contains samples with weak ferromagnetism and paramagnetic behaviors.

In brief, 77 K Mössbauer spectra were fitted, at least, with two different octahedral environments for Fe ions, and the results suggest that the presence of Cr ions decreases the B_hf_ value, but the difference between the two sextets of the Fe-substituted YCO compounds increases, except for the pure YFO, where only the sextet was required to have a good fit of the spectrum. One explanation for this decrease may be due to competing mechanisms between the antiferromagnetic interactions between Fe–Fe, Fe–Cr, and Cr–Cr exchanges and the DM interaction. Indeed, the asymmetric DM interaction is known to be the main interaction responsible for the WFM observed in YFO, where the antiferromagnetic coupling mechanism is due to superexchange interactions between the t3–O–t3 and e2–O–e2 orbitals, whereas for the YCO compound, the mechanism is a coupling to the t–e orbitals [22,23,24]. Therefore, we have in the Fe-substituted YCO samples a mixed exchange mechanism that is enhanced by the atomic disorder naturally present in our samples. It can be expected that due to the Fermi contact and the transferred magnetic field contribution to the total hyperfine magnetic field depend on the s electrons and the superposition of 3d, s, and p electrons, respectively, there is increasing competition of Fe environments as the iron concentration of the sample increases. An appropriate calculation using the mean field theory gave the relationship J_Fe–Fe_ > J_Fe–Cr_ > J_Cr–Cr_ [8].

## 4. Conclusions

In the present work, the structural and magnetic properties of the Fe-substituted perovskite series were studied in detail. Specifically, the average crystalline grain size of the YFeO_3_ compound, calculated using the harmonic spherical approach in a Rietveld refinement, was 84 (8) nm, a size where weak ferromagnetism can occur in this compound. In the Fe-substituted YFe_x_Cr_1−x_O_3_ compounds, X-ray and neutron diffraction patterns collected at RT and 2 K gave a linear increase in the lattice parameters of the orthorhombic structure with increasing Fe concentration, where the c-parameter had the most pronounced increase. This increase obviously translates into an increase in the volume cell and consequently a change in the magnetocrystalline anisotropy of the samples, i.e., a magnetic anisotropy that depends on the Fe concentration. Considering that X-ray and neutron diffraction showed only one crystalline phase for all samples, and the above results of lattice parameters that showed a gradual increase with increasing iron content, we can highlight that autocombustion is a useful method for the synthesis of pure YFeO_3_ with high stoichiometry. The 90 kOe *M(H)* curves taken at RT and 5 K for all Fe-substituted samples suggest the presence of spin reorientation and magnetization reversal phenomena associated with homogenous (Fe–Fe, Cr–Cr) and non-homogeneous pairs of 3d-ions (Fe–Cr). The dependence of the *H_C_*, *M_r_*, and *σ_sat_* with Fe concentration clearly showed the onset of WFM for x = 0.60–0.80 values. For x = 1.0, the high *H_C_* value of 46.7 kOe was calculated after subtracting the AFM contribution (linear contribution of the paramagnetic phase). This latter result implies that an enhancement in the WFM is achieved due to chemical inhomogeneity of the YFeO_3_ phase. Moreover, the values of *M_r_* and *σ_sat_* at 300 K are in agreement with the values commonly found in single crystals. The WFC and ZFC *M(T)* curves recorded at low (50 Oe) and high (1000 Oe) probe field analysis allowed the magnetic properties and global magnetic response of the spin reorientation process to be tuned. The high field *M(T)* curves allowed for accurate determination of *T_N_* values and showed a nonlinear dependence of *T_N_* on Fe concentration. The sample with x = 0.75 clearly exhibited a higher magnetic disorder, as corroborated by Mössbauer spectra recorded at 77 K and 300 K. The magnetization measurement performed in a low probe field of 50 Oe, for the sample with x = 0.50 showed a diamagnetic-like behavior near the compensation temperature of 245 K, where an inverse magnetization and the most intense remanence and saturation values at 300 K were found compared to the other samples. For the low Fe content (x = 0.25) and pure orthochromite samples, they showed paramagnetic-like behavior at RT, with a magnetic order only below 150 K. Mössbauer spectra allowed us to study the local Fe environment and visualize a weak enhanced ferromagnetism, due to a remarkable high canting angle (13°), estimated previously from neutron diffraction analysis, and its variation with Fe concentration. At least two octahedral Fe sites were identified in the non-pure Fe-substituted samples, whose evolution of the magnetic hyperfine field (B_hf_) can be explained using the results of mean field theory reported in the literature. The sample with x = 0.50 showed good magnetic properties and is a suitable candidate for further study.

## Figures and Tables

**Figure 1 nanomaterials-12-03516-f001:**
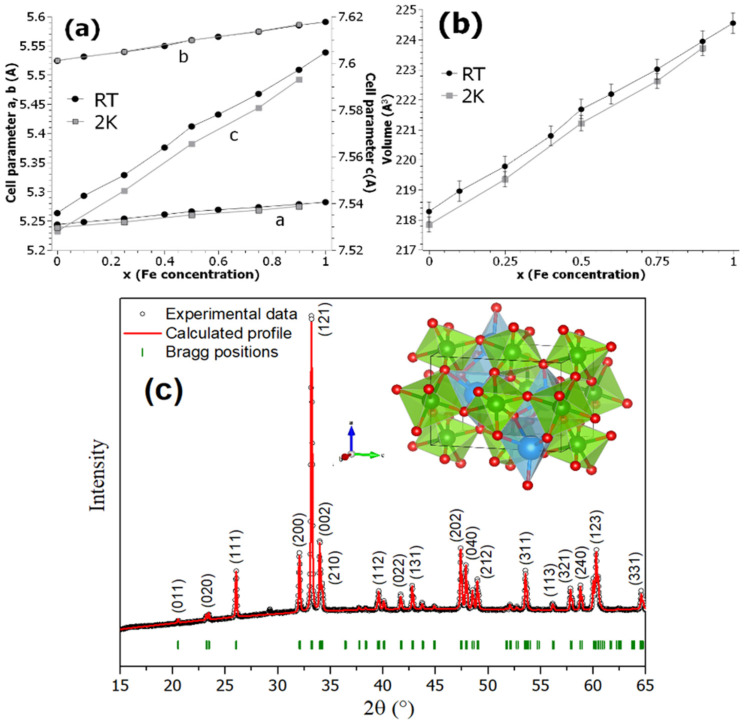
Cell parameter (**a**) and volume (**b**) variation with the x (Fe concentration) and Rietveld refinement XRD diffractogram (**c**) for the RS7 sample. The inset in (**c**) is the simulated structure obtained after refinement using VESTA (red spheres are oxygen atoms, white and blue are yttrium atoms, and green spheres are iron atoms). While blue, red, and green arrows indicate the a, b, and c crystallographic axes, respectively. Miller indexes are given between parentheses, black dots are the experimental data, the red line is the calculated diffractogram, and the vertical green lines are the Bragg’s diffraction position. Solid lines in (**a**,**b**) are guides for visualization.

**Figure 2 nanomaterials-12-03516-f002:**
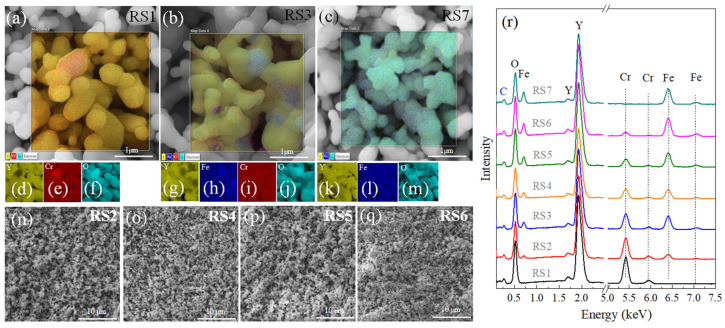
(**a**–**c**) SEM images for the RS1, RS3, and RS7 samples (bar length = 1 µm). The magnified area was performed in high resolution mode, and the elemental mapping area for each sample is given in (**d**–**m**) images. SEM images for the RS2, RS4, RS5, and RS6 samples (**n**–**q**) (bar length =10 µm). (**r**) The EDS for the RS1–RS7 samples.

**Figure 3 nanomaterials-12-03516-f003:**
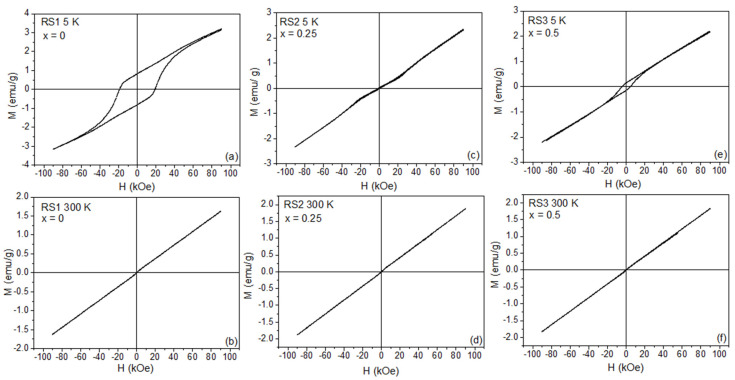
*M(H)* curves recorded for a maximum field of 90 kOe at 5 K (top) and 300 K (bottom) for the RSX samples. RS1 in (**a**,**b**), RS2 in (**c**,**d**), and RS3 in (**e**,**f**), respectively.

**Figure 4 nanomaterials-12-03516-f004:**
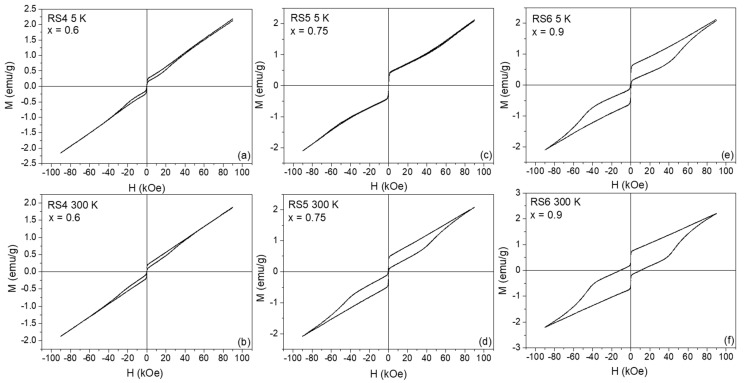
*M(H)* curves recorded for a maximum field of 90 kOe at 5 (top) and 300 K (bottom) for the RSX samples. RS4 in (**a**,**b**), RS5 in (**c**,**d**), and RS6 in (**e**,**f**), respectively.

**Figure 5 nanomaterials-12-03516-f005:**
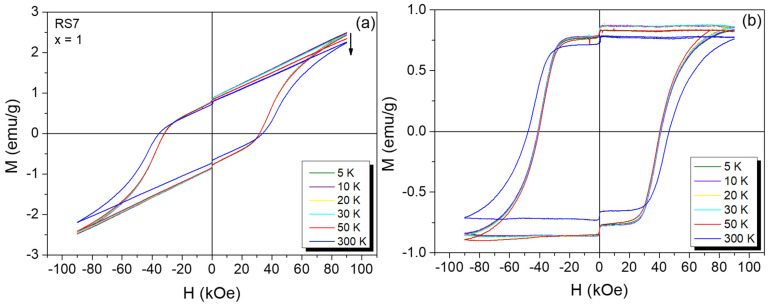
*M(H)* loops for the RS7 sample recorded at different temperatures (**a**). *M(H)* loops after the subtraction of the paramagnetic contribution of the AFM phase (**b**).

**Figure 6 nanomaterials-12-03516-f006:**
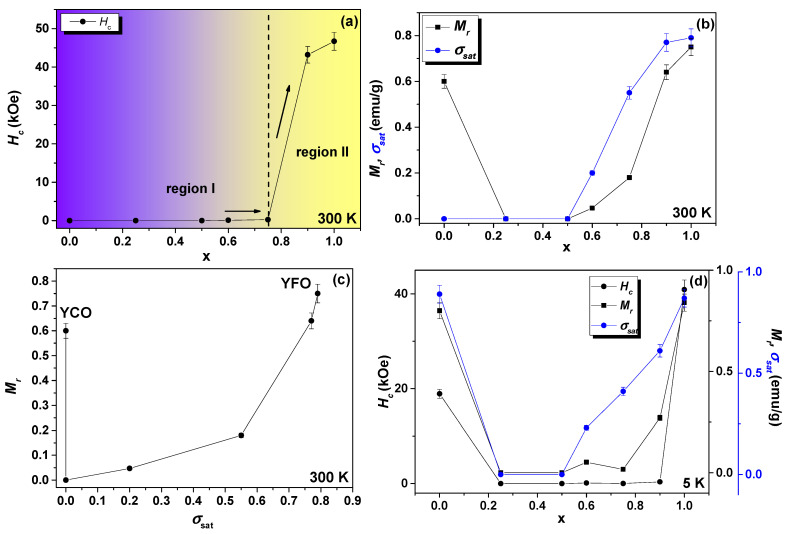
(**a**) Dependence of the *H_C_* (kOe) vs. x (Fe concentration) at 300 K. (**b**) Dependence of the *M_r_* and $\sigma_{sat}$ vs. x (Fe concentration) at 300 K. (**c**) *M_r_* vs. $\sigma_{sat}$ graph at 300 K. (**d**) *H_C_*, *M_r_*, and $\sigma_{sat}$ vs. (Fe concentration) at 5 K.

**Figure 7 nanomaterials-12-03516-f007:**
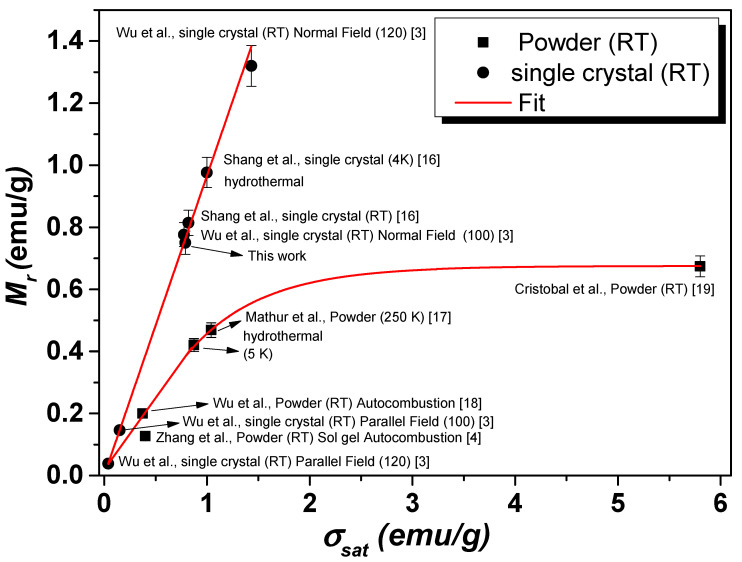
*M_r_* vs. $\sigma_{sat}$ relation built from data recorded at RT and reported in the literature for similar compounds. Comparison between powder and single crystal YFe_x_Cr_1−x_O_3_ [3,5,16,17,18,19].

**Figure 8 nanomaterials-12-03516-f008:**
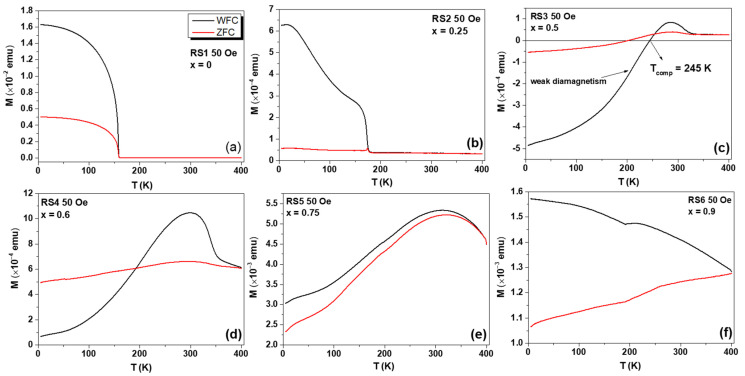
ZFC and WFC *M(T)* curves at a probe field of 50 Oe for the (**a**) RS1, (**b**) RS2, (**c**) RS3, (**d**) RS4, (**e**) RS5, and (**f**) RS6 samples.

**Figure 9 nanomaterials-12-03516-f009:**
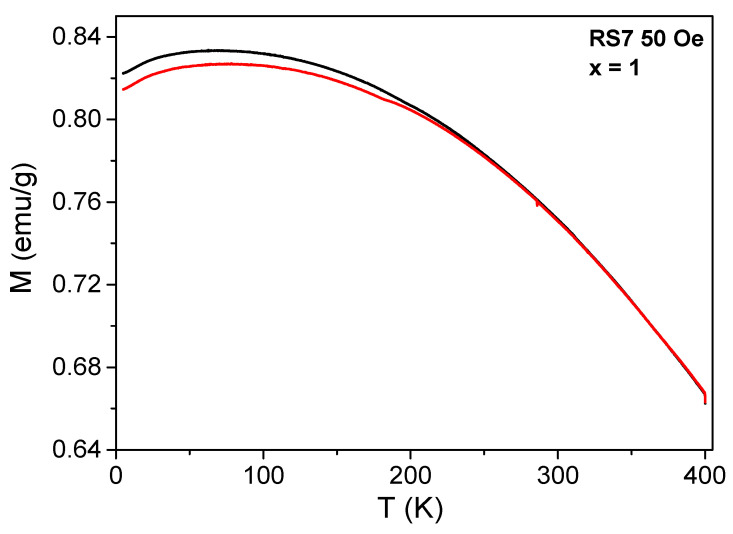
ZFC and WFC *M(T)* curves recorded for a probe field of 50 Oe for the RS7 sample. Black line indicates WFC and red line ZFC, respectively.

**Figure 10 nanomaterials-12-03516-f010:**
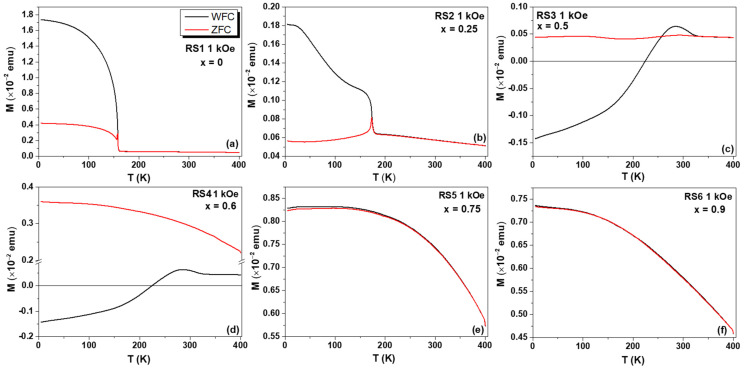
ZFC and WFC *M(T)* curves recorded at a probe field of 1000 Oe for the (**a**) RS1, (**b**) RS2, (**c**) RS3, (**d**) RS4, (**e**) RS5, and (**f**) RS6 samples.

**Figure 11 nanomaterials-12-03516-f011:**
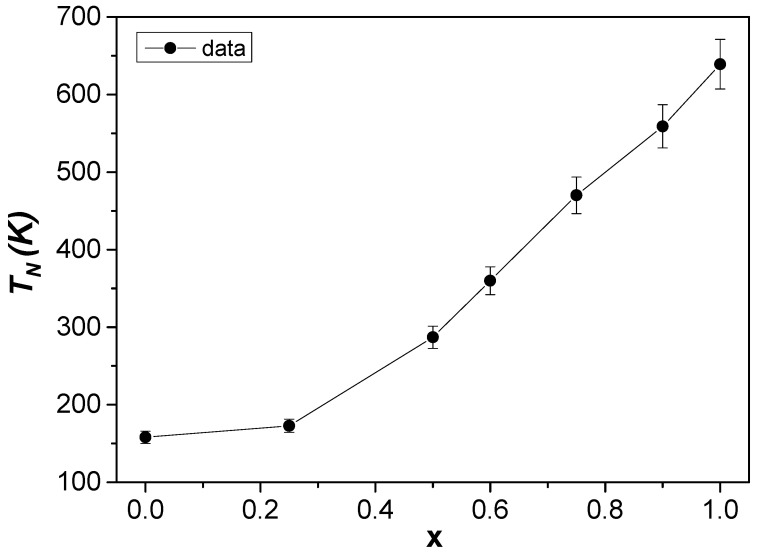
Néel temperature *T_N_* dependence on x (Fe concentration) estimated from 1000 Oe *M(T)* curves. The solid line passing by experimental points is only a guide for viewing.

**Figure 12 nanomaterials-12-03516-f012:**
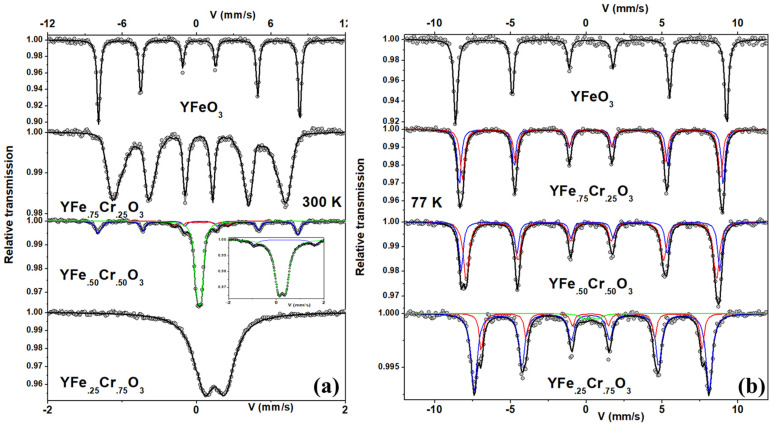
(**a**) RT Mössbauer measurements of the Fe-substituted YFe_x_Cr_1−x_O_3_ compounds. Samples with x = 1.0, 0.75, and 0.5 are measured with ±12 mm/s. The inset shows the same spectra for x = 0.5 but with a maximal velocity of ±2 mm/s, as in the case of the YFe_0.25_Cr_0.75_O_3_ sample (last spectrum). (**b**) Same sample measured at 77 K. The subspectra used to fit these spectra are also shown. The two sextets represent the two iron configurations in octahedral sites of the orthorhombic crystal structure of the perovskite.

**Table 1 nanomaterials-12-03516-t001:** Weight percentage composition of the elements found in all samples.

Sample	Y (% wt) ± 2	Cr (% wt) ± 2	Fe (% wt) ± 1	O (% wt) ± 1
RS1	52	31	-	17
RS2	51	25	9	15
RS3	47	17	23	14
RS4	51	14	20	15
RS5	51	9	25	16
RS6	51	4	30	15
RS7	51	-	33	16

**Table 2 nanomaterials-12-03516-t002:** Remanence (*M_r_*), coercive field (*H_C_*), and magnetic saturation (*σ_Sat_*) of the ferromagnetic component, and susceptibility of the antiferromagnetic component to the RSx samples with x = 1, 0.9, 0.75, and 0.60. For the samples with x = 0, 0.25, and 0.50, the values correspond to the ‘paramagnetic’ state.

x	T (K)	*M_r_* (emu/g) ± 0.05	*H_C_* (kOe) ± 0.5	*σ_Sat_* (emu/g) ± 0.05	Susc. (emu/g × Oe) ± 0.01
1.00	300	0.75	46.7	0.79	0.02
1.00	5	0.84	40.9	0.87	0.02
0.90	300	0.64	43.2	0.77	0.02
0.90	5	0.27	0.3	0.61	0.02
0.75	300	0.18	0.3	0.55	0.02
0.75	5	0.02	0.02 (5)	0.41	0.02
0.60	300	0.05	0.1	0.20	0.02
0.60	5	0.05	0.1	0.23	0.02
0.50	300	0	0	0	0.02
0.50	5	0	0	0	0.02
0.25	300	0	0	0	0.02
0.25	5	0	0	0	0.03
0.00	300	0	0	0	0.02
0.00	5	0.80	18.9	0.89	0.03

The remanence (*M_r_*) is calculated from *M_r_* = (*M_R_*_+_ + *M_R_*_−_)/2, where *M_R+_* and *M_R_*_−_ are the values of the upper and lower magnetization, respectively, when the magnetic field is zero. The coercive field (*H_C_*) is calculated from *H_C_* = (*H_C_*_+_ − *H_C_*_−_)/2, where *H_C+_* and *H_C−_* are the values of the right and left fields when magnetization is zero.

**Table 3 nanomaterials-12-03516-t003:** Refined values of the hyperfine parameters at given temperatures.

x	T (K)	IS (mm/s) ± 0.01	2ε or Δ (mm/s) ± 0.01	B_hf_ (T) ± 0.5	Absorption Area Ratio % *± 2*
1	300	0.37	0.00	50.1	100
	77	0.48	0.01	55.2	100
0.75	300	<0.38>	<0.04>	<40.7>	100
	77	0.47	0.03	53.8	49
		0.47	0.05	52.6	51
		<0.47>	<0.04>	<53.2>	
0.50	300	0.39	0.28		62
		0.40	−0.24	50.1	22
		0.46	−0.12	48.5	8
		0.35	0.15	13.3	8
	77	0.48	0.02	51.0	62
		0.48	−0.10	52.7	38
		<0.48>	<−0.02>	<51.7>	
0.25	300	0.36	0.28		100
	77	0.47	0.08	47.9	74
		0.47	0.08	45.0	23
		0.47	0.79		3

## Data Availability

The original data related to this research can be requested at any time from the corresponding author at the following email address: rsalazarr@uni.edu.pe.

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
