# Peer review of "Presence of Induced Weak Ferromagnetism in Fe-Substituted YFexCr1−xO3 Crystalline Compounds"

_nanomaterials, 2022, doi:10.3390/nano12193516_

Round 1

Reviewer 1 Report

The authors report a very interesting study on Fe substituted YFe xCr1-xO3. The results proof the presence of magnetization phenomena compatible with ferromagnetic ordering. Both Mossbauer and X-ray diffraction techniques were employed to support the magnetic characterization.  The sample characterization and results discussion is well presented, however additional comments and interpretation are necessary to explain the collapse in the coercivity  observed in Fig.3 and 4. Does this originate from Wasp Waisted hysteresis effects?

Author Response

Dear reviewer, thank you very much for your positive opinion about our manuscript. As explained in the main text the abrupt changes in the coercivity values are associated to spin reorientation of magnetic pairs yielding to the final YFO phase (lines 241-247). A wasp waisted effect will require of soft/hard coupling between different magnetic materials. In our case, we only have one phase raising to an exchange spring (x=1.0) and not to a wasp waisted like magnetic behavior.

Reviewer 2 Report

In this work, the structural and magnetic properties of the Fe-substituted YCO3 phase in the nanocrystalline regime are investigated in detail. Especially, the authors found the existence of weak ferromagnetic in the YFeO3 compounds. In addition, it was confirmed by the Mössbauer spectrometry performed at 300 K and 77 K and under external magnetic field. The results are interesting and novel. Moreover, the manuscript is well organized and a self-consistent explanation is given. The present work is worth for publication after minor revisions.

1. The analysis of the atomic percentage composition of the elements is not consistent with the stoichiometric YCrO3. Why the content of the O element is less than the theoretical one? Please give an explanation.

2. The authors claim that they conducted the XRD and neutron diffraction powder patterns at RT and 2 K. But these patterns are not shown. I suggested that the authors offer these patterns in the manuscript or the supplementary materials.

Author Response

Answer to the first question:
We thank the reviewer for their important comment. We have edited the Table 1 (line 214) by changing the atomic by weight percentage, see edited changes with yellow color. Errors are now in close relation to the theoretical one.

Answer to the second question:
Dear reviewer, as we mentioned in section 3.1. (line 145) these patterns were analyzed in reference 11. Hence, we believe it is not necessary to add them in this new work. By contrary, systematic analysis of the cell parameters are given in Figure 1(a) and (b).

Reviewer 3 Report

Here they report the synthesis and characterization of several compositions 21 across the series. Using a unique autocombustion technique, various compositions (x = 0.25, 0.50, 22 0.6, 0.75, 0.9, and 1) were synthesized as high quality nanocrystalline powders. To obtain both mi- 23 croscopic and atomic information on their structure and magnetism characterization was performed 24 using room temperature X-ray diffraction and energy dispersion analysis as well as temperature 25 dependent neutron diffraction, magnetometry and 57Fe Mössbauer spectrometry. Rietveld analysis 26 of the diffraction data revealed a crystallite size of 84 (1) nm for the YFeO3, while energy dispersion 27 analysis indicated compositions near to the nominal ones.

Author Response

Dear reviewer, thank you very much for your positive response.